# Exercise as a Therapeutic Strategy for Sarcopenia in Heart Failure: Insights into Underlying Mechanisms

**DOI:** 10.3390/cells9102284

**Published:** 2020-10-13

**Authors:** Jinkyung Cho, Youngju Choi, Pavol Sajgalik, Mi-Hyun No, Sang-Hyun Lee, Sujin Kim, Jun-Won Heo, Eun-Jeong Cho, Eunwook Chang, Ju-Hee Kang, Hyo-Bum Kwak, Dong-Ho Park

**Affiliations:** 1Institute of Sports & Arts Convergence (ISAC), Inha University, Incheon 22212, Korea; lovebuffalo@gmail.com (J.C.); choiyoungju0323@gmail.com (Y.C.); Jame0409@gmail.com (S.-H.L.); sujin2419@hanmail.net (S.K.); gjwnsdnjs03@naver.com (J.-W.H.); cejeong97@naver.com (E.-J.C.); change@inha.ac.kr (E.C.); johykang@inha.ac.kr (J.-H.K.); kwakhb@inha.ac.kr (H.-B.K.); 2Department of Cardiovascular Diseases, Mayo Clinic, Rochester MN 55905, USA; Sajgalik.Pavol@mayo.edu; 3Program in Biomedical Science & Engineering, Department of Kinesiology, Department of Biomedical Science, Inha University, Incheon 22212, Korea; 77nodaji@hanmail.net; 4Department of Pharmacology and Medicinal Toxicology Research Center, Inha University School of Medicine, Incheon 22212, Korea

**Keywords:** sarcopenia, heart failure, exercise, older adults

## Abstract

Sarcopenia, a syndrome commonly seen in elderly populations, is often characterized by a gradual loss of skeletal muscle, leading to the decline of muscle strength and physical performance. Growing evidence suggests that the prevalence of sarcopenia increases in patients with heart failure (HF), which is a dominant pathogenesis in the aging heart. HF causes diverse metabolic complications that may result in sarcopenia. Therefore, sarcopenia may act as a strong predictor of frailty, disability, and mortality associated with HF. Currently, standard treatments for slowing muscle loss in patients with HF are not available. Therefore, here, we review the pathophysiological mechanisms underlying sarcopenia in HF as well as current knowledge regarding the beneficial effects of exercise on sarcopenia in HF and related mechanisms, including hormonal changes, myostatin, oxidative stress, inflammation, apoptosis, autophagy, the ubiquitin-proteasome system, and insulin resistance.

## 1. Introduction

Heart failure (HF) is an age-related geriatric health issue, defined as reduced cardiac output due to myocardial dysfunction. Over 5.8 million people in the US and over 23 million people worldwide suffer from this syndrome [1]. The prevalence of HF increases sharply in patients aged ≥65 years, and the disease is strongly associated with a number of pathophysiological complications, including pulmonary hypertension, renal disease, vascular dysfunction, and stroke [2,3]. Additionally, patients with HF often demonstrate a reduced exercise capacity that restricts daily activity and mobility [4]. There are several causes of exercise intolerance in HF patients, including systemic impairment of blood perfusion, blood flow, oxygen supply, and diffusivity of blood to skeletal muscle tissues [4]. Such chronic impairments of the cardiovascular system, along with limited physical activity, may cause quantitative and qualitative loss of skeletal muscle and accelerate the onset of sarcopenia [5].

Sarcopenia is an age-related geriatric syndrome that is characterized by qualitative and quantitative loss of skeletal muscle [6]. The prevalence of sarcopenia gradually increases with age, with ~30% of older adults, aged 59–86 years, being affected by this syndrome [6]. Growing evidence suggests that sarcopenia is commonly associated with HF as a comorbidity that synergistically influences a reduction in physiological function and physical mobility [7,8]. Moreover, the prevalence of sarcopenia in HF patients is approximately 20% higher than that in the healthy elderly population [9]. Sarcopenia is considered a salient health issue associated with HF and a factor that plays an important role in the maintenance of mobility and quality of life [7]. However, given the lack of standard treatment options, attenuating the progression of sarcopenia in HF poses a serious challenge. Accordingly, this review will discuss sarcopenia-related pathophysiology in HF and the therapeutic effect of exercise as a potential treatment strategy for attenuating sarcopenia progression associated with HF.

## 2. Potential Mechanisms of Sarcopenia in Heart Failure

Due to the general physiological degeneration that accompanies advance in age, gradual loss of muscle is commonly seen in older adults, even in the absence of sarcopenia. However, this pathophysiological phenomenon appears to be facilitated by HF [9]. Haykowsky et al. [10] reported that older patients with HF who presented with preserved ejection fractions showed significantly lower percentages of total lean muscle mass and leg muscle mass relative to those in healthy age-matched controls. Additionally, a decrease in the fat-free mass index, which reflects muscle mass, appeared to be associated with an increased risk for negative cardiac health outcomes in patients with HF [10,11]. Thus, muscle loss combined with HF may accelerate pathophysiological changes and reduce mobility in elderly patients with HF.

Sarcopenia is clinically defined by the European Working Group on Sarcopenia in older People using the following criteria: a low appendicular lean mass (AML) adjusted for height (AML/height^2^ <7.0 kg/m^2^ for men and <5.5 kg/m^2^ for women), low muscle strength (handgrip strength <27 kg for men and <16 kg for women), and/or poor physical function (gait speed ≤0.8 m/s) [6]. Although the above-stated clinical evaluation appears to focus on quantitative muscle loss, the pathophysiology of sarcopenia involves metabolic changes, particularly accelerated catabolism [12]. Accordingly, there is growing interest in the qualitative loss of muscle, the broad physiological changes resulting from which include altered levels of anabolic hormones, myostatin, oxidative stress, inflammation, apoptosis, autophagy, ubiquitin-proteasome system activity, and insulin resistance [12,13,14]. These changes lead to an imbalance between muscle protein synthesis and degradation, which, in turn, results in the wasting of skeletal muscle, leading to cardiometabolic and functional abnormalities in patients with HF.

### 2.1. Hormonal Changes

Decreased anabolic hormone levels have been investigated for association with skeletal muscle wasting. Age-related decreases in anabolic hormones, including growth hormones (GHs), insulin-like growth factor-1 (IGF-1), and testosterone, contribute to muscle loss by disrupting muscle metabolism [15,16,17]. Reduced GH and IGF-1 levels in HF are associated with impaired cardiac performance and lower exercise tolerance, which possibly contribute to muscle loss, and individuals with HF often demonstrate serum GH and IGF-1 levels that are lower than those of healthy age-matched controls [16]. Moreover, a randomized controlled study indicated that long-term GH replacement therapy improved the cardiac function of GH-deficient patients with chronic HF, as evidenced by enhanced peak oxygen consumption (peak VO_2_) values, left ventricular ejection fractions, and left ventricular end-systolic volumes [16]. Recently, an animal study conducted by Brioche et al. [18] demonstrated that administering GH alleviated sarcopenia, as shown by improvements in muscle protein synthesis and mitochondrial biogenesis. These findings suggest that decreased levels of GH and IGF-1 are associated with sarcopenia and HF.

Testosterone is the primary male sex anabolic hormone. Testosterone levels in older healthy people are a subject of considerable debate. A number of cross-sectional studies have reported that testosterone levels decrease with advancing age and that age-related decrease in testosterone levels in older adults is associated with a corresponding decrease in muscle mass and muscle strength [19]. Storer et al. [20] reported that older men who underwent a 3-year-long testosterone treatment showed improved muscle performance and physical function, including stairclimbing power and muscle mass, compared with older men in the placebo group. Testosterone deficiency is often observed in HF patients. It has been shown that low level of testosterone in HF is associated with a worse prognosis and increased mortality [21,22]. Further, low testosterone levels are associated with reduced peak VO_2_ in men with HF [23]. Thus, decreasing testosterone levels may contribute to skeletal muscle atrophy and dysfunction in HF.

### 2.2. Myostatin 

Myostatin, a member of the transforming growth factor-β superfamily, is released mainly by skeletal muscle and, to a lesser extent, by the heart and adipose tissue [24]. Muscle growth, which is mediated by the downregulation of the Akt pathway and upregulation of the ubiquitin-proteasome system, is inversely regulated by myostatin [25]. The serum myostatin levels of HF patients were elevated compared with those of age-matched healthy controls [26]. Lenk et al. [27] reported that, in HF patients, myostatin mRNA and protein levels in skeletal muscle were augmented compared with healthy subjects. These findings suggested that myostatin may play a role in HF-related muscle wasting.

### 2.3. Oxidative Stress

Oxidative stress occurs due to imbalances between the production of reactive oxygen species (ROS) and antioxidant defenses. ROS are mainly produced by mitochondria, and excessive production of ROS contributes to reduced mitochondrial function and oxidative capacity [28]. Upregulated mitochondrial apoptosis signaling induced by oxidative damage may accelerate the atrophying of skeletal muscle in HF patients [29]. In addition, exercise intolerance associated with HF may be increased due to oxidative damage related to endothelial dysfunction [30]. Thus, appropriate regulation of oxidative stress may exert positive effects against muscle atrophy in HF.

Nicotinamide adenine dinucleotide phosphate (NADPH) oxidase, which comprises seven isoforms (i.e., NOX1-5, Duox1, and Duox2), is a source of ROS production [31]. This enzymatic complex appears to be elevated in patients with HF [32]. A previous study demonstrated that elevation of myocardial NADPH oxidase activity in HF is associated with the modulation of redox-sensitive signaling pathways, such as those involving mitogen-activated protein (MAP) kinases (ERK1/2, ERK5, p38, MAPK) [33]. Further, NADPH oxidase overactivity, following increased ROS production, is also associated with nuclear factor kappa B (NF-κB) signaling and the ubiquitin-proteasome system in skeletal muscle, both of which may participate in the pathogenesis of sarcopenia [34].

### 2.4. Inflammation

The serum concentrations of pro-inflammatory markers, such as interleukin-6 (IL-6), C-reactive protein, and tumor necrosis factor (TNF), are known to be elevated in HF patients [35,36]. Increased inflammation is associated with disease severity and adverse prognosis [37]. Levine et al. [35] reported that HF-related increases in serum TNF-α levels are associated with skeletal muscle wasting. Additionally, studies have suggested that an increase in pro-inflammatory cytokines during skeletal muscle wasting is indicative of the role played by these cytokines in the development of sarcopenia, which, in turn, demonstrates the involvement of these cytokines in muscle loss in HF patients [36,38]. In vivo data have been supported by in vitro studies involving muscle cells, which indicated that treatment with TNF-α inhibited myoblast differentiation and limited the regenerative response of satellite cells to muscle damage by inducing NF-κB [39]. Moreover, TNF-α/NF-κB signaling elevates ROS production via mitochondrial electron transport chain, resulting in stimulation of muscle protein degradation [40]. Therefore, chronic low-level systemic inflammation may be an important contributing factor in HF-related muscle catabolism.

### 2.5. Apoptosis

Myonuclear apoptosis, which is characterized by the absence of myonuclei and condensation of sarcoplasm, contributes to fiber atrophy rather than cell death, resulting in skeletal muscle atrophy [41]. Activation of the apoptotic signaling pathway leads to protein degradation via the ubiquitin-proteasome system [42]. Adams et al. [43] reported that myonuclear apoptosis, which is frequently found in HF patients compared with age-matched healthy control subjects, was linked to exercise intolerance.

### 2.6. Autophagy

Autophagy is an intracellular degradation process that removes misfolded proteins and damaged organelles [44]. Activation of the autophagic pathway in skeletal muscle plays a crucial role in maintaining protein homeostasis via several signaling pathways, including phosphatidylinositol-3-kinase (PI3K), myostatin, proteasome, and autophagy-lysosome pathways [44]. Any imbalance between these multiple signaling pathways during aging results in loss of muscle mass and function. Recent findings have reported that impaired autophagy in skeletal muscle is seen in HF animal models, which may contribute to skeletal muscle damage and degeneration [45,46,47,48]. Thus, autophagy signaling must be properly regulated to maintain skeletal muscle quality and quantity in HF patients.

### 2.7. Ubiquitin-Proteasome System 

Increased protein degradation is recognized as one of the mechanisms underlying muscle loss in HF patients. The ubiquitin-proteasome system is known to regulate intracellular protein degradation in skeletal muscle cells. This degradation process is mediated by the activation of ubiquitin ligases (E3), such as atrogin-1 and muscle RING-finger protein-1 (MuRF-1), which polyubiquitinate proteins in order to mark them for degradation by the 26S proteasome [49]. A study reported that elevation of atrogin-1 and MuRF-1 levels is associated with muscle atrophy and promotes the pathological progression of sarcopenia [49]. Interestingly, the levels of E3 ligases in the diaphragm and quadriceps of an animal with chronic HF model were found to be elevated [50], suggesting that activation of the ubiquitin-proteasome system contributes to cardiac and peripheral muscle dysfunction in HF patients. 

### 2.8. Insulin Resistance 

During aging, a decrease in muscle mass combined with an increase in intramuscular fat content interferes with insulin-mediated glucose usage, causing insulin resistance [51]. Such metabolic abnormalities associated with skeletal muscle are also related to HF [52,53]. A clinical study showed that quadriceps muscle strength in both patients with HF and healthy controls was positively correlated with the insulin sensitivity index [52,53]. These findings suggest that insulin resistance is pathologically linked to skeletal muscle loss. Mechanistically, insulin resistance impairs insulin/IGF-1 signaling, which regulates phosphatidylinositol-3-kinases/Akt signaling and attenuates protein regeneration [54]. Promotion of muscle catabolism and impairment of protein synthesis and muscle wasting by these activities indicate an association between insulin resistance and HF-related skeletal muscle wasting.

## 3. Exercise Intolerance and Sarcopenia in HF

Exercise intolerance, a major symptom of HF, is often utilized to predict disease severity and mortality. To quantify exercise capacity, peak VO_2_ is measured during a cardiopulmonary exercise test. This test is commonly accepted as the gold-standard for such assessments [55]. A decrease in peak VO_2_ in HF may be largely due to complications in the central region, including reduced carbon monoxide (CO), impaired respiratory gas exchange, and insufficient blood flow [56]. In addition, systemic oxygen delivery to locomotor muscles is significantly reduced during exercise because of impaired endothelium-dependent vasodilation [57], muscle capillarization [58], systemic inflammation, oxygen unloading [59], and oxidative stress [60]. These cardiovascular impairments mainly limit exercise capacity in HF. However, emerging evidence supports the postulate that muscle waste/sarcopenia contributes to impaired exercise capacity in HF [61], based on the link between sarcopenia and multiple pathophysiological changes taking place in skeletal muscle, which lead to locomotor muscle dysfunction during exercise.

To the best of our knowledge, studies that are currently investigating the direct impact of skeletal muscle wasting on exercise capacity in HF are either very few or non-existent. However, previous studies have found that exercise capacity is related to skeletal muscle mass and that reduced exercise capacity in older cohorts is associated with age-related decline in muscle mass [62]. Longitudinal studies using muscle biopsies have indicated that the capillary-to-fiber ratio in the skeletal muscles of older adults was significantly lower than that in the skeletal muscles of younger adults, demonstrating the association between exercise intolerance and limited oxygen supply [51]. Lower skeletal muscle capillarization was also found to be associated with a reduction in the exercise capacity of older adults, which was again found to be due to reduced oxygen supply [63]. Haykowsky et al. [10] demonstrated that the peak VO_2_ level for a given lean body mass was lower in HF patients than in healthy controls, implying that skeletal muscle abnormalities may contribute to exercise intolerance in HF patients. Recently, Weiss et al. [64] reported that HF patients with exercise intolerance and high fatigability exhibited significantly faster rates of exercise-induced decline in skeletal muscle high-energy phosphate and reduced maximal oxidative capacity than HF with low fatigability. These findings indicate that skeletal muscle metabolism may contribute to exercise intolerance in HF, suggesting an avenue for developing a strategy to improve exercise capacity.

## 4. Effects of Exercise on HF-Related Sarcopenia

Currently, there is no standard treatment for loss of muscle mass and function in HF patients. However, growing evidence suggests that lifestyle-modifiable factors, particularly regular exercise training, may help improve the muscular function and fitness of HF patients. The beneficial effects of exercise on the cardiovascular system have been reported frequently [65,66,67]. Since potential mechanisms underlying sarcopenia and HF appear to be shared, exercise training may simultaneously improve cardiac and skeletal muscle functions associated with HF-related sarcopenia, although the processes associated with such potential improvements remain unclear. A conceptual overview of the potential effects of exercise on HF-related sarcopenia is shown in Figure 1.

### 4.1. Physical Activity

Epidemiologic studies have demonstrated that appropriate amounts of physical activity may enable the preservation of muscle-related functions, including gait, speed, balance, and daily activities, in addition to reducing the risk of age-related diseases, such as sarcopenia and cardiovascular diseases [68,69,70]. Additionally, the Framingham Study, which involved 1142 older adults, demonstrated that individuals with higher physical activity displayed a 15–56% lower risk for HF [71], whereas a cross-sectional study by Ribeiro Santos et al. [72] found that less active older adults engaged in occupational and locomotion domains showed a higher risk of sarcopenia. Overall, although the relationship between HF-related sarcopenia and physical activity remains insufficiently studied, existing evidence suggests that adequate physical activity may prevent HF-related muscle loss.

### 4.2. Exercise Training 

Exercise training is a well-known strategy that improves maximal oxygen consumption (VO_2max_) as well as muscle mass [73]. Although the impact of exercise training on HF patients is not fully understood, it is clear that exercise training enables muscle mass to be effectively maintained and reduces muscle abnormalities in HF. Notably, exercise training increases the mass and activity of skeletal muscle mitochondria, thereby enhancing exercise tolerance in HF. Numerous human and rodent studies have demonstrated the beneficial effects of exercise training on skeletal muscles in HF (Table 1 and Table 2).

#### 4.2.1. Aerobic Exercise Training 

HF-induced muscle abnormalities that contribute to skeletal muscle atrophy result from an imbalance between protein synthesis and degradation, which is a function of reduced anabolic gene expression and increased catabolic gene expression. On the contrary, clinical studies revealed that aerobic exercise training (AET) enhanced peak VO_2_, increased anabolic gene expression (i.e., IGF-1), reduced catabolic gene expression (i.e., MuRF-1, myostatin, and TNF-α), and improved the oxidative capacity of skeletal muscle [27,74,75,76,77,78]. Similar results were observed in animal models. For example, AET upregulated IGF-1 levels through activated Akt and ERK1/2 signaling while reducing forkhead box O3 (FoxO3) mRNA levels in the skeletal muscle of myocardial infarction model [84,86]. An animal study by Bacurau et al. [80] demonstrated that 8 weeks of AET prevented muscle loss by activating Akt and mammalian target of rapamycin (mTOR) and subsequently increasing the phosphorylation of p70-S6 kinase 1 (p70S6K). Although the exact mechanisms underlying the AET-mediated activation of protein synthesis in human HF-induced muscle atrophy are not clearly understood, it is assumed that AET stimulates protein synthesis via IGF-1 and the related signaling pathways such as Akt/mTOR and Akt/FoxO3.

Moreover, AET may exert beneficial effects by reducing the expression of catabolic genes such as myostatin [27]. Deletion of the myostatin gene in heart tissues prevented skeletal muscle atrophy in animal HF models [87]. Lenk et al. [81] showed that 4 weeks of AET resulted in significant reduction in myostatin expression levels in both the skeletal muscle and myocardium in an animal model of HF. The decrease in myostatin expression levels in skeletal muscle depends on the p38 MAPK-dependent pathway, which involves NF-κB [81]. However, in contrast to changes in skeletal muscle, no significant differences were observed between serum myostatin levels following AET [27,88]. Although myostatin is expressed in both local tissues (i.e., skeletal muscle, heart, and adipose tissue) and systemic pools [89], AET may modulate myostatin levels in the skeletal muscle of HF patients via localized action.

Histological analysis of skeletal muscle from patients with HF have shown the changes of muscle fiber type from type Ι oxidative fibers to type Ⅱ glycolytic fibers, which leads to skeletal muscle mitochondrial dysfunction [90]. The findings of many studies indicate that mitochondrial dysfunction in skeletal muscle contributes to exercise intolerance in HF as estimated by peak VO_2_ and may therefore be considered as a potential therapeutic target for this condition. Molina et al. [91] examined the content, dynamics, proteins, and oxidative capacity of skeletal muscle mitochondria using a vastus lateralis skeletal muscle biopsy from an HF patient. The levels of mitochondrial fusion factors, mitofusin 2 (Mfn2) and porin, as well as citrate synthase activity, were lower in HF patients than those in age-matched healthy control subjects [91]. However, mitochondrial dysfunction may be partially normalized by exercise training in HF [83,92]. Our group showed that acute AET-induced upregulation of PGC-1α levels regulated the signaling pathway modulating mitochondrial biogenesis and protected muscles from oxidative stress, proteolysis, and inflammation [93]. Similar findings were reported by Souza et al. [82] in an animal model of HF. Ten weeks of AET suppressed the deterioration of muscle loss, and maintained the PGC-1α levels in the skeletal muscle. Furthermore, AET-induced anti-oxidative effects may be associated with redox-sensitive proteins, such as NADPH oxidase. Using animal HF models, Cunha et al. [76,79] demonstrated that 8 weeks of AET suppressed the progression of HF pathology as well as skeletal muscle atrophy. In this study, AET regulated proteolysis signaling by reducing the protein levels of Atrogin-1/muscle atrophy F-box (MAFbx) and E3α. AET also modulated the redox balance in the membrane-enriched fraction of plantaris by downregulating Nox2 and p47phox protein levels as well as NADPH oxidase hyperactivity. These findings suggested that AET may partially reverse certain features of skeletal muscle abnormalities, such as mitochondrial function, in HF patients.

#### 4.2.2. Resistance Exercise Training 

Some studies have reported that although resistance training (RT) leads to improved muscle strength and function, AET provides greater benefits by improving cardiorespiratory fitness [85,94,95,96]. Accordingly, Gomes et al. [85] compared the effects of aerobic and resistance exercise on the physical capacity and skeletal muscle oxidative stress in HF rats. The results of this study indicated that AET and RT increased the aerobic capacity and strength gain of HF rats respectively, regardless of changes caused by cardiac remodeling. Thus, RT can be expected to play a crucial role in improving muscle strength and function in HF patients. For example, Esposito et al. [78] examined the effects of 8 weeks of knee extensor exercise training on skeletal muscle function in HF patients. Compared with the baseline, RT induced a 1.5-fold increase in 50% of maximum work rate along with muscle capillary density, fiber cross-section area, and percentage area of type I fiber [78]. Cai et al. [84] reported that 4 weeks of RT inhibited muscle atrophy by downregulating the mRNA levels of MuRF-1 and atrogin-1, decreased ROS, and upregulated the IGF-1/Akt/ERK signaling pathway in soleus muscle. These data indicate that the regulation of protein synthesis-related genes in skeletal muscle may be modulated in response to RT in HF.

#### 4.2.3. Combined Exercise Training

Current guidelines recommend regular physical activity for HF patients to alleviate cardiac and muscular dysfunction [3]. As far as exercise types are concerned, aerobic exercise has been suggested as the primary mode due to beneficial effects exerted on anabolic hormones, antioxidant status, and inflammatory markers, as well as on muscle and cardiovascular function. Since resistance exercise is considered relatively unsafe for HF patients, the beneficial effects of exercise have mostly been attributed to aerobic exercise [94]. Jakovljevic et al. [97] studied the effects of both aerobic and resistance exercise on HF patients and concluded that both increased cardiac pumping capabilities and physical functional capacities. Moreover, they showed that while peak VO_2_ improvements were more closely associated with aerobic exercise, muscle mass was best maintained via RT [97]. All findings considered, aerobic exercise combined with RT appears to be effective in preventing muscle loss associated with HF.

All European nations and South America guidelines recommend both AET and RT for improving the muscle mass and function of HF patients [98]. Thus, a training procedure that involves a combination of two types of exercise training strategies may be most beneficial for improving physical function in HF [99]. Accordingly, using vastus lateralis muscle biopsy samples from older adults, Irving et al. [100] showed that oxidative capacity and expression of mitochondrial proteins (i.e., Mitochondrial Oxidative Phosphorylation (OXPHOS) protein and citrate synthase) and transcription factors (i.e., PGC-1α, Sirtuin 3 (SIRT3), and mitochondrial transcription factor A (TFAM)) after 8 weeks of combined training were superior to those after either AET or RT alone. These findings suggest that exercise intervention, especially via combined exercise training, may attenuate abnormalities in muscle quantity and quality, thereby minimizing deleterious peripheral conditions, such as muscle loss and endothelial dysfunction, in HF patients. However, more standardized clinical trials are needed in the future to explore optimal exercise prescriptions that may be used to improve cardiac and skeletal muscle function.

## 5. Conclusions

Patients with HF exhibit a loss of cardiovascular function and skeletal muscle quantity and quality due to sarcopenia, which can lead to poor prognosis. Although the mechanisms underlying the direct interaction between HF and sarcopenia remain unclear, those underlying skeletal muscle wasting in HF patients appear to be associated with anabolic hormones, myostatin, oxidative stress, inflammation, apoptosis, autophagy, the ubiquitin-proteasome system, and insulin resistance. Currently, standard methods that may be used to treat sarcopenia in HF patients have not been established. However, exercise training represents a non-pharmacological method that shows potential for attenuating skeletal muscle wasting in these patients. This review attempted to highlight potential mechanisms via which exercise plays a role in decreasing muscle loss in patients with HF. Further studies are needed to provide additional molecular and clinical evidences concerning effective exercise protocols, particularly in terms of type, intensity, and volume of exercise required.

## Figures and Tables

**Figure 1 cells-09-02284-f001:**
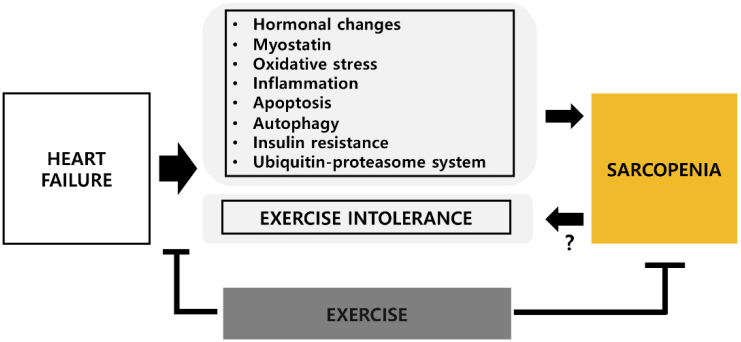
Schematic overview of the potential beneficial effects of exercise on heart failure-related sarcopenia.

**Table 1 cells-09-02284-t001:** Beneficial effects of exercise training in patients with heart failure.

Authors	Study Design	Subjects	Exercise Type	Beneficial Effects of Exercise
Lenk et al. [27]	RCT	Chronic HF patients	Type: aerobic exercise (bicycle ergometer) Duration: 12 weeks Frequency: daily Time: 20–30 min/day	In skeletal muscle ⇓ Catabolic gene expression (Myostatin) In serum ⇔ Myostatin level
Lelyavina et al. [74]	RCT	HF patients	Type: aerobic exercise (walking) Duration: 12 weeks Frequency: 4–5 times/week	⇑ Peak VO_2_ (mL/kg/min) ⇑ Left ventricular ejection fraction (%) ⇑ Exercise tolerance In skeletal muscle ⇓ Fiber diameter, endomysium thickness
Gielen et al. [75]	RCT	Chronic HF patients	Type: aerobic exercise (bicycle ergometer) Duration: 4 weeks Frequency: 4 times/week Time: 20 min/day	⇑ Peak VO_2_ (mL/kg/min) In vastus lateralis muscle biopsies ⇓ Catabolic gene expression (*MuRF-1*) ⇑ Anabolic gene expression (*IGF-1*) ⇓ Inflammatory gene expression (*TNF-*α)
Cunha et al. [76]	RCT	HF patients	Type: aerobic exercise (walking) Duration: 12 weeks Frequency: 3 times/week Time: 50 min/week	⇑ Peak VO_2_ (mL/kg/min) In skeletal muscle ⇓ Proteasome activity
William et al. [77]	RCT	Chronic HF patients	Type: resistance training (upper and lower body) Duration: 11 weeks Frequency: 3 session/week	⇑ Peak VO_2_ (mL/kg/min) ⇑ Lactate threshold (W) ⇑ Peak lactate (mmol/L) ⇑ Capillary density In skeletal muscle ⇑ Oxidative capacity (CS, HAD enzyme activity) ⇑ MAPR (mmol ATP/min/kg)
Esposito et al. [78]	RCT	HF patients with reduced ejection fraction	Type: resistance training (knee extensor exercise) Duration: 8 weeks Frequency: 3 times/week	In skeletal muscle ⇑ Type 1 fiber ⇑ Mitochondria volume ⇔ *VEGF* mRNA

RCT, randomized controlled trial; HF, heart failure; MuRF-1, muscle RING-finger protein-1; IGF-1, insulin-like growth factor-1; TNF-α, tumor necrosis factor-α; CS, citrate synthase; HAD, beta-hydroxyacyl CoA-dehydrogenase; MAPR, mitochondrial ATP production rate; VEGF, vascular endothelial growth factor; ⇑, upregulation; ⇓, downregulation; ⇔, no significant changes.

**Table 2 cells-09-02284-t002:** Beneficial effects of exercise training in animal model with heart failure.

Authors	Subjects	Exercise Type	Beneficial Effects of Exercise
Cunha et al. [76]	α_2A_/α_2C_ARKO mice	Type: Treadmill running Duration: 8 weeks Intensity: moderate intensity Frequency: 5 days/week	⇑ Exercise performance In plantaris muscle ⇑ Cross area of muscle ⇓ *Atrogin-1/MAFbx* and *E3α* mRNA levels
Cunha et al. [79]	LAD-ligation rat	Type: Treadmill running Duration: 8 weeks Intensity: Moderate intensity Frequency: 5 days/week	⇑ Exercise performance ⇑ Type 1 fiber percentage ⇓ Serum *TNFα* In plantaris muscle ⇓ ROS ⇓ *NOX2, p47phox* and NADPH oxidase ⇓ *NF-kB* and *p38 MAPK*
Bacurau et al. [80]	α_2A_/α_2C_ARKO mice	Type: Treadmill running Duration: 8 weeks Intensity: Moderate intensity Frequency: 5 days/week	⇑ Exercise performance ⇑ Soleus atrophy In soleus muscle ⇑ *IGF-1/Akt/mTOR* signaling, ⇓ Proteasome activity (*p4E-BP1/4E-BP1, p-p70S6K/p70S6K*)
Lenk et al. [81]	LAD-ligation rat	Type: Treadmill running Duration: 4 weeks Intensity: 30 m/min Frequency: 5 days/week	In gastrocnemius muscle ⇓ Myostatin expression In muscle cell ⇑ Myostatin via *TNFα/p38MAPK/NFkB* signaling pathway
Souza et al. [82]	Aortic stenosis surgery rat	Type: Treadmill running Duration: 10 weeks Intensity: 15 m/min Frequency: 5 days/week	⇑ Serum *IGF-1* In soleus and plantaris muscle ⇑ Cross area of muscle ⇑ CS activity ⇑ *PGC1α* ⇓ *FOXO1* mRNA
Moreira et al. [83]	LAD-ligation rat	Type: Treadmill running Duration: 8 weeks Intensity: moderate intensity (60% VO_2_max vs. 85% VO_2_max) Frequency: 5 days/week	⇑ Exercise performance In soleus muscle ⇑ *PDK* mRNA ⇑ CS activity ⇑ Glycogen content ⇓ *Atrogin-1, MuRF1* mRNA In plantaris muscle ⇑ CS activity ⇑ Glycogen content ⇑ Hexokinase ⇓ *Atrogin-1* mRNA
Cai et al. [84]	Myocardial infarction surgery rat	Type: Resistance exercise (Ladder climbing) Duration: 3 sessions/day, 4 weeks Intensity: moderate intensity Frequency: 5 days/week	In soleus muscle ⇓ ROS ⇓ *Atrogin-1* and *MuRF-1* ⇑ *IGF-1/Akt/ERK*
Gomes et al. [85]	LAD-ligation rat	Type: Treadmill running vs. resistance exercise (ladder climbing) Duration: 12 weeks Intensity: moderate intensity Frequency: 3 days/week	Treadmill running ⇑ Maximum exercise capacity Ladder climbing ⇑ Maximum carrying load In gastrocnemius muscle ⇓ Lipid hydroperoxide ⇑ Glutathione peroxidase activity ⇑ Superoxide dismutase activity

ROS, reactive oxygen species; α2A/α2CARKO, α2A- and α2C-adrenergic receptors knock out mice; LAD-ligation rat, left anterior descending artery-ligation rat; TNF-α, tumor necrosis factor-α; NF-kB, nuclear factor-κB; MAPK, mitogen-activated protein kinases; IGF-1, insulin like growth factor-1; mTOR, mammalian target of rapamycin; NOX2, NADPH oxidase 2; PGC1α, peroxisome proliferator-activated receptor gamma coactivator 1-alpha; FOXO1, Forkhead Box O1; PDK1, 3-phosphoinositide-dependent protein kinase-1; MuRF-1, muscle RING-finger protein-1; CS, citrate synthase; ERK, extracellular signal-regulated protein kinase; ⇑, upregulation; ⇓, downregulation; ⇔, no significant changes.

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
