# Peer review of "Exercise as a Therapeutic Strategy for Sarcopenia in Heart Failure: Insights into Underlying Mechanisms"

_cells, 2020, doi:10.3390/cells9102284_

Round 1

Reviewer 1 Report

Overall, this MS is descriptively revised. However, I think that the paragraph concerning autophagy (Line161-168) should be revised.

Although the author presented only one article of De Meyer et al. [Ref. 45] to introduce the relationship between autophagy-dependent signaling and HF, this is review and not original article. Thus, the author should present more appropriate original articles in this field.

For example.

Bowen TS et al. (2018) Effects of endurance training on detrimental structural, cellular, and functional alterations in skeletal muscles of heart failure with preserved ejection fraction. J Card Fail. 24(9): 603-613. doi: 10.1016/j.cardfail.2018.08.009.

Fujita N et al. (2015)  Time course of ubiquitin-proteasome and macroautophagy-lysosome pathways in skeletal muscle in rats with heart failure. Biomed Res. 36(6): 383-392.

doi: 10.2220/biomedres.36.383.

Janning PR et al. (2014) Autophagy signaling in skeletal muscle of infarcted rats. PLoS One 9(1): e85820.

doi: 10.1371/journal.pone.0085820. eCollection 2014.

Author Response

Point-by-point Response:

Reviewer #1

Q1) Overall, this MS is descriptively revised. However, I think that the paragraph concerning autophagy (Line161-168) should be revised.

Although the author presented only one article of De Meyer et al. [Ref. 45] to introduce the relationship between autophagy-dependent signaling and HF, this is review and not original article. Thus, the author should present more appropriate original articles in this field.

For example.

Bowen TS et al. (2018) Effects of endurance training on detrimental structural, cellular, and functional alterations in skeletal muscles of heart failure with preserved ejection fraction. J Card Fail. 24(9): 603-613. doi: 10.1016/j.cardfail.2018.08.009.

Fujita N et al. (2015)  Time course of ubiquitin-proteasome and macroautophagy-lysosome pathways in skeletal muscle in rats with heart failure. Biomed Res. 36(6): 383-392.

doi: 10.2220/biomedres.36.383.

Janning PR et al. (2014) Autophagy signaling in skeletal muscle of infarcted rats. PLoS One 9(1): e85820.

doi: 10.1371/journal.pone.0085820. eCollection 2014.

ANS#1) We appreciated a valuable review of the reviewer. We absolutely agree with your comment. Now, we clarified by the new sentence in the 2.6. Autophagy section. And we newly added references that the reviewer recommended in the reference section.

“Any imbalance between these multiple signaling pathways during aging results in loss of muscle mass and function. Recent findings have reported that impaired autophagy in skeletal muscle is seen in HF animal models, which may contribute to skeletal muscle damage and degeneration [45-48]. Thus, autophagy signaling must be properly regulated to maintain skeletal muscle quality and quantity in HF patients.”

  1. Jannig, P.R.; Moreira, J.B.; Bechara, L.R.; Bozi, L.H.; Bacurau, A.V.; Monteiro, A.W.; Dourado, P.M.; Wisloff, U.; Brum, P.C. Autophagy signaling in skeletal muscle of infarcted rats. PLoS One 2014, 9, e85820, doi:10.1371/journal.pone.0085820.
  2. Fujita, N.; Fujino, H.; Sakamoto, H.; Takegaki, J.; Deie, M. Time course of ubiquitin-proteasome and macroautophagy-lysosome pathways in skeletal muscle in rats with heart failure. Biomed Res 2015, 36, 383-392, doi:10.2220/biomedres.36.383.
  3. Bowen, T.S.; Herz, C.; Rolim, N.P.L.; Berre, A.O.; Halle, M.; Kricke, A.; Linke, A.; da Silva, G.J.; Wisloff, U.; Adams, V. Effects of endurance training on detrimental structural, cellular, and functional alterations in skeletal muscles of heart failure with preserved ejection fraction. J Card Fail 2018, 24, 603-613, doi:10.1016/j.cardfail.2018.08.009.

Reviewer 2 Report

I would like to thank the authors for the revisions which have been made to the manuscript which are perfect.

Author Response

Thanks for the good evaluation!

This manuscript is a resubmission of an earlier submission. The following is a list of the peer review reports and author responses from that submission.

Round 1

Reviewer 1 Report

Very good review article. Authors did a great job in summarizing the literature. 

Reviewer 2 Report

This is a review concerning muscle wasting following cardiac failure. As such it has extensively covered the field of this muscle wasting induced by HF, its possible causes and ways of combating this muscle loss. However, I find the manuscript a little confusing at time since there seems to be a confusion between . Sarcopenia - the normal age related loss of muscle mass and quality begining as early as 40 years of age and accelerating rapidly after 70-80 years of age and the additional loss of muscle mass frequently refered to as muscle wasting/ muscle atrophy triggered by heart failure which can even lead to cachexia - whole body wasting. This additional muscle loss which is due to many factors including reduced physical activity and oxidative stress due to HF in the elderly will superimpose upon the natural age related loss of muscle/ sarcopenia.

Please could the author try to make this clearer in the text.

In several places throughout the text sarcopenia should be replaced by muscle wasting or muscle atrophy.

A good example is in the coblclusion where you should just state that patients with HF exhibit a loss of cadiovascular function and skeletal muscle mass and quality.

Also there have been a number of recent reviews such as those by von Haehlling 2018, Springer et al 2017, Suzuki et al 2018 which unless I am mistaked have not been cited.

Reviewer 3 Report

Sarcopenia, a common syndrome in elderly populations, is often characterized by a gradual loss of skeletal muscle and results in functional decline and loss of independence. Growing evidence suggests that the prevalence of sarcopenia increases in patients with heart failure (HF), which is a dominant pathogenesis in the aging heart. HF causes diverse metabolic complications and in turn this may result in sarcopenia. Therefore, sarcopenia can be a strong predictor of frailty, disability and mortality in HF. There remains no standard treatment for slowing muscle loss in this population. The authors reviewed the pathophysiological mechanisms of sarcopenia in HF and discussed the use of exercise as a therapeutic strategy for sarcopenia. Furthermore, they highlight current knowledge regarding the beneficial effects of exercise on sarcopenia and its underlying mechanisms, including hormonal changes, myostatin, oxidative stress, inflammation, insulin resistance, and the ubiquitin–proteasome system. At glance, this review article may be intriguing. However, there are several important mistakes in this article.

The authors write the potential mechanism of sarcopenia in heart failure. This is terribly bad.

  1. The content of myostatin (2.2) is very old and poor probably due to no reference of recent articles in this field.
  2. The content of ubiquitin-proteasome system (2.5) is terribly humorous. Now, the contribution of ubiquitin-proteasome system for sarcopenia is completely denied. Recently, the defect of autophagy-dependent system is mainly considered to contribute the enhancement of sarcopenic symptom. The authors should read recent articles in this field with taking more time.
  3. The author’s group is amateur in this field although possessing only one original article [Ref. 82]. Unfortunately, almost all the references are very old-fashioned. Although the authors use easily the word of “recent”, the referred articles are 2013, 2014, 2015. Unfortunately, I think that the word of the recent article means after 2017 or 2018.